# Comparison of Vascular Density Changes After Cataract Surgery in Diabetic Patients with and Without Pseudoexfoliation Syndrome Using Optical Coherence Tomography Angiography

**DOI:** 10.3390/biomedicines13040908

**Published:** 2025-04-08

**Authors:** Lelde Svjascenkova, Arturs Zemitis, Janis Gredzens, Guna Laganovska

**Affiliations:** 1Department of Doctoral Studies, Riga Stradins University, Dzirciema Street 16, LV-1007 Riga, Latvia; arturs.zemitis@gmail.com (A.Z.); glaganovska@ml.lv (G.L.); 2Department of Ophthalmology, Pauls Stradins Clinical University Hospital, Pilsonu Street 13, LV-1002 Riga, Latvia; 3Laboratory of Statistical Research and Data Analysis, University of Latvia, LV-1586 Riga, Latvia; gredzensjanisjg@gmail.com

**Keywords:** type 2 diabetes mellitus, pseudoexfoliation syndrome, optical coherence tomography angiography, diabetic retinopathy, phacoemulsification cataract surgery, vascular density

## Abstract

**Background:** This study aims to evaluate changes in the central retina in patients with type 2 diabetes mellitus (T2DM) undergoing uncomplicated small incision cataract surgery with or without pseudoexfoliation syndrome (PEXS) using optical coherence tomography angiography (OCTA). **Methods:** In this prospective, longitudinal study, 67 T2DM patients underwent cataract surgery. Twelve of them had PEXS. All parameters were measured at two time points. Macular 3 × 3 mm and 6 × 6 mm OCTA images were obtained. All data were analyzed using R statistical software (version 4.4.1). **Results:** Parafoveal vascular density (VD) in the superior capillary plexus and the deep capillary plexus increased in the non-PEX group. There was an increase in VD in perifoveal vascular density in the deep capillary plexus in both groups. Three months after cataract surgery, changes in perifoveal vascular density in the deep capillary plexus increased in both groups and were significant. **Conclusion:** Perifoveal vascular density in the deep capillary plexus showed a significant increase in VD, regardless of the presence of PEXS. Parafoveal VD in the deep and superficial capillary plexuses appeared to be sensitive primarily in non-PEXS patients, with a notable increase observed in these areas three months after surgery.

## 1. Introduction

Pseudoexfoliation syndrome (PEXS) is a systemic disorder characterized by the accumulation of fibrillar extracellular material in various tissues, including the eye. It is a well-established risk factor for glaucoma and is associated with significant ocular and systemic vascular abnormalities [1,2]. In the eye, PEXS affects the anterior segment and has been linked to reduced vascular density (VD) in the peripapillary region, which is a critical factor in glaucomatous damage [1]. However, while the relationship between PEXS and glaucoma has been extensively studied, less attention has been given to its impact on the central retinal vasculature, particularly in the context of comorbid conditions, such as diabetes mellitus (DM) [3].

Diabetes mellitus, a systemic metabolic disorder, is known to cause microvascular changes in the retina, leading to diabetic retinopathy and other vision-threatening complications. Both PEXS and DM share common pathways of vascular dysfunction, including endothelial damage and impaired blood flow regulation, which may exacerbate retinal vascular changes [3,4]. Interestingly, the relationship between PEXS and DM remains controversial, with some studies suggesting a higher prevalence of PEXS in diabetic patients, while others report an inverse association, particularly in older populations [5]. This discrepancy underscores the need for further research into the interplay between these two conditions and their combined effects on retinal vasculature.

Optical coherence tomography angiography (OCTA) has emerged as a powerful, non-invasive imaging tool for evaluating retinal vascular density in both superficial and deep capillary plexuses. Unlike traditional fluorescein angiography, OCTA allows for the detailed visualization of the deep capillary plexus, providing new insights into microvascular changes associated with various ocular and systemic conditions [6]. Recent studies have demonstrated reduced macular VD in eyes with PEXS, suggesting that microvascular injury may precede structural damage in these patients [6]. However, the impact of cataract surgery on retinal vascular density in diabetic patients with and without PEXS remains poorly understood.

Cataract surgery, one of the most commonly performed surgical procedures worldwide, has been shown to influence retinal hemodynamics and vascular density. While the surgery is generally safe, its effects on the retinal vasculature may differ in patients with systemic conditions, such as DM and PEXS, which are associated with compromised vascular integrity. Understanding these differences is crucial for optimizing postoperative care and improving visual outcomes in these high-risk populations.

The primary objective of this study was to investigate and compare changes in retinal vascular density before and after uncomplicated cataract surgery in diabetic patients with and without PEXS using OCTA. By examining differences in the avascular zone, vascular density, and other microvascular parameters, we aimed to elucidate the distinct retinal changes associated with PEXS in the context of diabetes. Our findings have important clinical implications, as they may help identify patients at higher risk of postoperative vascular complications and guide the development of targeted management strategies to preserve visual function.

## 2. Materials and Methods

This prospective, longitudinal, non-experimental study included patients diagnosed with type 2 diabetes. In this study, 67 eyes of 67 patients were included in the analysis. All participants underwent phacoemulsification cataract surgery with intraocular lens implantation at Pauls Stradins Clinical University Hospital in Latvia between October 2020 and September 2024. Ethical approval for the study was granted by the Riga Stradins University Ethics Committee on 13 August 2020 (decision number NR 6-1/09/4). The study adhered to the principles outlined in the Declaration of Helsinki, and all participants provided written informed consent prior to undergoing surgery. The inclusion criteria for patients undergoing cataract surgery required participants to be at least 18 years of age with a confirmed diagnosis of T2DM as evidenced by serological testing that included serum blood glucose levels and glycated hemoglobin (HbA1C). Exclusion criteria included a history of ocular trauma or the presence of other ocular diseases affecting the macula, such as age-related macular degeneration, glaucoma, advanced cataract, corneal opacification, or previous vitrectomy. Participants were stratified into groups based on the presence or absence of PEXS. Additionally, patients were categorized into two groups based on diabetic retinopathy status: those with no detectable diabetic retinopathy changes and those with non-proliferative diabetic retinopathy (NPDR). Additionally, all of them were divided into diabetic retinopathy stages using a 5-stage classification: 1—no DR; 2—mild NPDR; 3—moderate NPDR; 4—severe NPDR; 5—proliferative diabetic retinopathy. In the NPDR group, 23 patients had stage 2 (mild) NPDR, 6 had moderate NPDR, and 5 had severe NPDR. Fifteen patients had diabetic maculopathy with or without significant diabetic macular edema. Five patients had received anti-VEGF injections within the past year (not 6 months prior to cataract surgery), and 12 had laser photocoagulation (LFC), mostly locally in the periphery and/or macula, depending on diabetic retinopathy stage and diabetic maculopathy. Additionally, LFC was not performed at least 6 months prior to cataract surgery for more precise study results.

All patients underwent complete ophthalmic examinations, which included best-corrected visual acuity (BCVA) measurement, intraocular pressure measurement, slit lamp examination, including fundus examination, optical coherence tomography, and OCTA, to evaluate the macula at baseline (before cataract surgery) and three months after surgery. All patients included in the study underwent phacoemulsification cataract surgery with intraocular lens implantation. The surgical procedures were conducted by multiple experienced surgeons, with no intraoperative complications reported. Postoperative management consisted of a standardized regimen of topical dexamethasone and levofloxacin administered over a two-week period. To minimize potential confounding factors, patients with a history of prior intravitreal injections or laser photocoagulation were included only if their most recent treatment had been performed at least three months before the cataract surgery.

All parameters for microvascular evaluation were measured using an OCTA device (Optovue RTVue XR 100 Avanti Edition Software, Version 2015.0, Optovue, Inc., Fremont, CA, USA) [7]. Macular 3 × 3 mm OCTA images were obtained to visualize the foveal avascular zone (FAZ), and 6 × 6 mm images were obtained to assess vascular density. Area wideness for central and macular analyses was chosen based on previous research in diabetic patients [8]. Central retinal thickness was measured from the inner limiting membrane to the retinal pigment epithelium. Mean para- and perifoveal vessel densities were measured in the superficial capillary plexus and deep capillary plexus. Additionally, we measured the FAZ and its perimeter. All parameters were calculated automatically, and only 7/10 quality scans were included for further analysis. The existence of PEX was determined during slit lamp biomicroscopy and gonioscopy examination. Comparisons of basic parameters between groups were performed using both conventional parametric and nonparametric tests, depending on the underlying variable distributions that were tested for normality using the Shapiro–Wilk test. Pearson’s Chi-square test was used to assess the existence of PEX, independent of gender. An independent two-sample t-test and the Wilcoxon rank sum test with two-sided and one-sided alternatives were used to compare the results in each group separately at different time points. Values are expressed as the median and *p*-values. A paired t-test and paired Wilcoxon rank sum test with two-sided and one-sided alternatives in each direction were used to compare the results between groups before and three months after surgery. All the data were analyzed using R (version 4.4.1) [9]. A *p*-value < 0.05 was considered statistically significant.

## 3. Results

A total of 67 patients were included in the analysis, comprising 49 women (73%) and 18 men (27%). The age of the participants ranged from 57 to 92 years, with a mean age of 73.0 years (±6.71) and a median age of 74 years as shown in Figure 1.

Participants were stratified into groups based on the presence of PEXS and DR status. Most of the cohort comprised non-PEXS patients (n = 55), while the PEXS subgroup was smaller (n = 12), reflecting the clinical rarity of concurrent PEXS and T2DM. Despite the limited PEXS sample size, statistically significant differences in retinal VD were observed between groups, suggesting robust biological effects detectable even in this niche population. These findings suggest that PEXS may have distinct effects on retinal microvasculature in diabetic patients, independent of sex or DR severity, as shown in Table 1. Furthermore, the lack of association between PEXS and sex/DR type reinforces its potential role as an independent modulator of postoperative vascular remodeling.

Most comparisons between PEX and non-PEX groups show no statistically significant differences, except for BCVA at baseline, which was initially 29% lower in PEX patients (*p* < 0.05, Table 2). Additionally, patients with PEX had a 15% higher central retinal thickness (CRT) three months after cataract surgery (second follow-up; *p* < 0.05).

There was no decrease in BCVA values for any of the PEXS patients; however, three instances of lower BCVA values were observed in non-PEX patients as shown in Figure 2.

More significant results were obtained by comparing values within groups at two time points, before and after the surgery.

Table 3 shows multiple measurements with statistically significant changes for both PEX and non-PEX subjects within the groups at two time points. All significant results are shown in bold. Abbreviations: BCVA, best-corrected visual acuity; FAZ, foveal avascular zone; CRT, central retinal thickness; FAZPER, foveal avascular zone perimeter; PARSCP, parafoveal superficial capillary plexus; PARDCP, parafoveal deep capillary plexus; PERDCP, perifoveal deep capillary plexus; PERSCP, perifoveal superficial capillary plexus.

The most notable change in relative terms is the change in BCVA; it increased by more than twofold for both groups compared to the initial measurements, with mean increases of 108% and 174% and median increases of 167% and 117% for the non-PEX and PEX groups, respectively (Figure 3).

CRT also increased in both groups. CRT increased more (twice as much) in the PEXS group, although the baseline was higher than in the non-PEX group. The increase in both the PEX and non-PEX groups was statistically significant (*p* < 0.05 for both), making a 3% increase in the non-PEX group (Figure 4).

A significant change during follow-up was observed in parafoveal vascular density in the superior capillary plexus (PARSCP), making a 6% increase in the non-PEX group (*p* < 0.05), as well as a 2% increase in parafoveal density in the deep capillary plexus (PARDCP) and the non-PEX group (*p* < 0.05). There was a 12% increase in perifoveal vascular density in the deep capillary plexus (PERDCP) (*p* < 0.05) in both PEX and non-PEX groups (Figure 5).

Another measure that increased in both groups and showed statistically significant changes is PERDCP (*p* < 0.05 for both non-PEX and PEX groups).

The changes within groups, as indicated in Table 3 and described above, are shown in the figures below.

Figure 3 shows that the overall increase in BCVA was more pronounced in the PEX group, primarily due to lower baseline measurements.

Figure 4 shows that, although the change was not large, it was consistent for almost all subjects with follow-up measurements, except for a single PEX patient, whose CRT value decreased (C plot).

**Figure 4 biomedicines-13-00908-f004:**
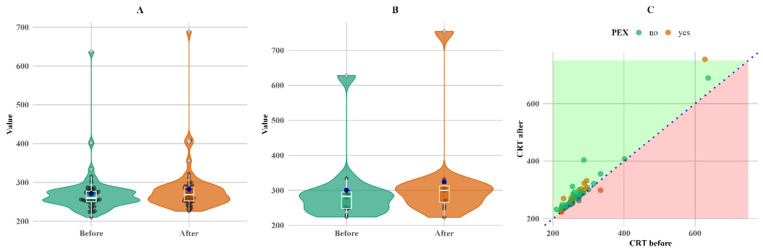
Within-group comparison of CRT. (**A**) Non-PEX; (**B**) PEX; and (**C**) all before and after values. Means (blue dots) and medians (middle line on box plots) in (**A**) non-PEX, (**B**) PEX, and (**C**) all before and after values.

Figure 5 shows that PERDCP increased equally in both groups, with six patients (one in the PEX group and five in the non-PEX group) having lower values.

**Figure 5 biomedicines-13-00908-f005:**
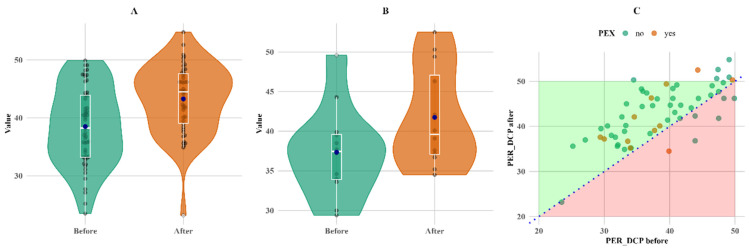
Within-group comparison of PER DCP. (**A**) Non-PEX; (**B**) PEX; and (**C**) all before and after values. Means (blue dots) and medians (middle line on box plots) in (**A**) non-PEX, (**B**) PEX, and (**C**) all before and after values.

## 4. Discussion

In our study, only 12 patients (18%) were diagnosed with PEXS, while the majority (55 patients, 82%) did not have PEXS. Most participants were within the 70–80 age range, which is consistent with U. Rumelaitiene et al.’s study, where the highest age of PEXS patients in all age groups was more than 75 years [10], as shown in Figure 1. Previous studies have reported a lower prevalence of PEXS among individuals with diabetes mellitus compared to non-diabetic populations. Balasopoulou et al.’s study found that only 6.7% of the 500 patients had PEXS, with half being diabetic and half being nondiabetic [7]. These findings are consistent with those of our study. According to M. Yu et al.’s meta-analysis, there is no correlation between DM and PEXS in general, although inverse relationships between the two may exist in elderly populations. The prevalence of PEXS ranges from 0.2% to 30%, depending on the study population examined and the detection method applied [5], which is also consistent with our study. Table 1 indicates the distribution of patients with and without PEXS. Seven patients with PEXS had no diabetic retinopathy, and five had non-proliferative diabetic retinopathy, resulting in a nonsignificant distribution (*p* > 0.05). PEXS is independent of gender, meaning that the probability of having PEXS is not significantly different between males and females; however, PEXS is more common in older people [10]. It is also independent of DR type (*p* < 0.05), as seen in Table 1.

This study has several limitations. First, the small sample size of the PEXS subgroup (*n* = 12) may limit the generalizability of our findings, despite the statistically significant differences observed. While this reflects the clinical rarity of PEXS in patients with type 2 diabetes mellitus, larger cohorts are needed to confirm the distinct effects of PEXS on postoperative retinal vascular remodeling. Second, the follow-up period was restricted to three months, which precludes the assessment of long-term microvascular changes or clinical outcomes, such as diabetic macular edema. Nevertheless, our results provide critical early insights into surgically induced alterations in vascular density in diabetic patients, particularly in the understudied PEXS population. Future multicenter studies with extended follow-up durations are warranted to validate these findings.

The relatively short follow-up period may not be sufficient to determine whether the observed changes in VD persist over time or revert to baseline levels. While our findings demonstrate significant VD alterations in the early postoperative period, these changes may stabilize or resolve with longer follow-up. For example, transient changes in retinal hemodynamics and vascular remodeling following cataract surgery could contribute to the observed VD differences, which may not reflect long-term outcomes. Future studies with extended follow-up periods are needed to evaluate the persistence of these changes and their potential implications for postoperative management and visual outcomes.

The most significant differences between the PEXS and non-PEXS groups were observed by comparing baseline and follow-up BCVA measurements. In both groups, BCVA improved significantly three months after surgery; however, the PEXS group demonstrated a markedly higher relative mean improvement of 174% (*p* < 0.05), as detailed in Table 3. This highlights a more pronounced postoperative visual recovery in patients with PEXS compared to those without the condition. These results raise questions about whether removing fibrillar material from the anterior chamber significantly improves visual acuity. Other studies, for example, Desinayak et al. [11] and Ilveskoski et al. [12] have found better visual outcomes after cataract surgery in groups without PEXS. Additionally, most of the fibrillar material is evacuated during phacoemulsification; however, some of the material may remain or reform, as shown by a gonioscopy study comparing pre- and post-cataract surgery [13].

Another notable observation is that there was no decrease in BCVA values for any of the PEXS patients; the fact that three instances of lower BCVA values were observed in non-PEX patients could be explained by an increase in CRT after surgery, causing a noticeable decrease in BCVA.

Significant damage to the retinal and choroidal vascular systems is seen in eyes with PEXS [10]. The superficial capillary plexus was more significantly affected in eyes with PEXS compared to those without PEXS.

Furthermore, in our study, CRT, FAZ, and VD were measured in different macular areas and depths. We found that patients with PEXS had 15% higher CRT three months after cataract surgery (*p* < 0.05). However, for the follow-up CRT, this increase may have been caused by missing follow-up data for seven non-PEX patients; thus, this result may be misleading. Despite the missing data, the results correspond with other research studies, such as that by Ilveskoski et al., which found that CRT after cataract surgery was significantly higher in PEXS patients [12]. Additionally, Pasaoglu I et al.’s study on the analysis of CRT asymmetry in the eyes using OCTA found that vessel density in the deep capillary plexus was lower in eyes with PEXS [14].

There were no significant changes in FAZ within or between the groups. However, E. Cinar et al.’s study found that FAZ was significantly enlarged in both the superficial and deep layers in PEXS eyes compared to control eyes [15]. Chae et al. and Kocaturk et al. also found that FAZ circularity was significantly lower in eyes without PEXS [16,17]; however, the circularity was quite normal, about 0.29 mm^2^ in the PEXS and non-PEXS groups, and did not change significantly three months post-operation. However, it is important to consider that all the patients in our study have T2DM, which inherently induces a certain degree of ischemia.

PERDCP increased in both groups and showed statistically significant changes (*p* < 0.05) for both non-PEX and PEX groups after surgery. As in previous studies, the deep capillary plexus is more sensitive to the inflammatory and hemodynamic changes that occur during and after cataract surgery [18]. Perifoveal vascular density values measured at baseline were not significantly different between the groups; however, in some studies, e.g., Düzova et al., parafoveal vascular density was considerably greater in the control group compared to glaucoma patients [19].

Other notable changes include the increase in parafoveal vascular density in the superficial capillary plexus and PARDCP plexus only in the non-PEX (PARSCP and PARDCP, *p* < 0.05) group three months after surgery, which could mean that, for non-PEX patients, these layers are still sensitive to intraocular hemodynamic changes that occur during surgery. E. Cinar et al. and Chae et al. found the superficial retinal vascular plexus plays a vital function in supplying the ganglion cell layer and is intimately associated with it; the studies also suggested that the superficial retinal layer and ganglion cell layer were considerably thinner in PEX eyes [15,16], making it similar to our study. Malyszczak et al. also found that T2DM patients have lower SCP compared to non-DM patients [20].

Comparing results at baseline showed no significant changes in vascular density between groups. Different results were noted in Zia Sultan Pradhan et al.’s study, where OCTA findings showed lower vascular density in patients with PEXS and PEX glaucoma (PEXG). In eyes with PEXS, peripapillary retinal nerve fiber layer thickness, parafoveal ganglion cell complex thickness, and parafoveal vessel density (VD) were reduced compared to controls [21]. Based on the study by E. Cinar et al., PEXS eyes exhibited significantly lower flow in the total, parafoveal, and foveal areas of the deep capillary plexus compared to control eyes (*p* < 0.05 for all). Significant reductions in VD values were observed across all regions in eyes with PEXG compared to both fellow eyes and control eyes [15]. Pradhan Zia et al. observed lower vascular density in the PEXG group compared to the POAG group, suggesting a potentially higher prevalence of vascular damage in PEXG patients [19]. Most of the studies mentioned analyzed glaucoma patients and their vascular densities, and evidence comparing VD in PEXS and T2DM patients is lacking.

While this study focused on the impact of PEXS and diabetes on retinal vascular density, it is important to acknowledge that other systemic conditions, such as hypertension and cardiovascular disease, may also contribute to vascular changes in these patients. Hypertension, for example, is known to cause endothelial dysfunction and reduce retinal blood flow, which could exacerbate the vascular alterations observed in PEXS and diabetic patients [22]. Similarly, cardiovascular disease may further compromise vascular integrity, leading to reduced retinal perfusion and an increased risk of ischemic damage [23].

While the grade of cataract, surgically induced astigmatism, and biometry precision are well-established factors influencing preoperative and postoperative BCVA, our findings suggest that PEXS may also play a role in visual outcomes. Interestingly, recent studies have proposed that ferroptosis, a form of regulated cell death characterized by iron-dependent lipid peroxidation, may underlie the extracellular matrix abnormalities and vascular dysfunction seen in PEXS [24]. This mechanism could provide a unifying explanation for the systemic and ocular manifestations of PEXS, including its association with comorbidities such as hypertension and cardiovascular disease. Further research is needed to explore the role of ferroptosis in PEXS and its potential implications for retinal vascular health.

## 5. Conclusions

In this study, we observed significant differences in retinal VD and clinical outcomes between diabetic patients with and without PEXS following cataract surgery. Patients with PEXS exhibited lower baseline BCVA, which improved significantly after surgery, suggesting that cataract removal may benefit visual function in this population. However, the increase in CRT in PEXS patients postoperatively highlights their predisposition to complications, such as macular edema.

Notably, the parafoveal VD in the deep and superficial capillary plexuses showed distinct responses to cataract surgery, with significant increases observed primarily in non-PEXS patients. These findings underscore the importance of considering PEXS as a factor influencing retinal vascular changes in diabetic patients.

Given the systemic and ocular implications of PEXS, including its association with cardiovascular and ischemic risk factors, multidisciplinary care involving general practitioners, endocrinologists, and ophthalmologists is essential for managing these patients. Routine eye examinations, including OCTA and gonioscopy, should be prioritized to detect and manage retinal vascular complications and secondary glaucoma promptly.

## Figures and Tables

**Figure 1 biomedicines-13-00908-f001:**
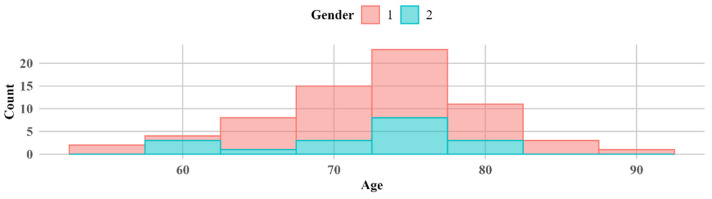
Histogram of age by gender. 1—women; 2—men.

**Figure 2 biomedicines-13-00908-f002:**
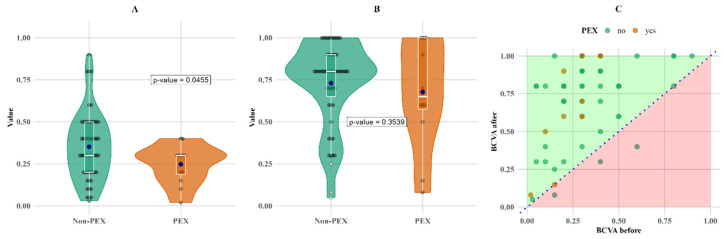
BCVA comparisons between groups. BCVA initial value distribution comparison between PEX and non-PEX subjects (**A**), follow-up BCVA value distribution comparison between patients with and without PEX (**B**) as well as before–after BCVA value scatter plot grouped by PEX (**C**). Means (blue dots) and medians (middle line on box plots) in plots (**A**,**B**) for initial and follow-up measurements. Abbreviations: PEX, pseudoexfoliation; BCVA, best-corrected visual acuity.

**Figure 3 biomedicines-13-00908-f003:**
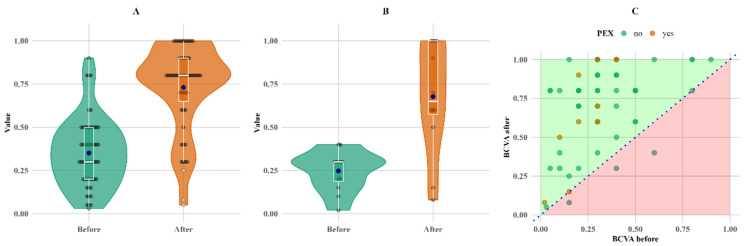
Within-group comparison of BCVA. Means (blue dots) and medians (middle line on box plots) in (**A**) non-PEX; (**B**) PEX; and (**C**) all before and after values.

**Table 1 biomedicines-13-00908-t001:** PEX frequency by gender and DR type.

Variable		Without PEX n = 55 (82%)	With PEX n = 12 (18%)	Total n = 67 (100%)	*p*
Sex	Female	41 (84%)	8 (16%)	49 (73%)	0.8427
	Male	14 (78%)	4 (22%)	18 (27%)	
DR type	no DR	25 (78%)	7 (22%)	32 (48%)	0.6239
	NPDR	30 (86%)	5 (14%)	35 (52%)	

Abbreviations: PEX, pseudoexfoliation; DR, diabetic retinopathy; NPDR, non-proliferative diabetic retinopathy.

**Table 2 biomedicines-13-00908-t002:** PEX and non-PEX group variable value comparisons.

Variable	Non-PEX N	PEX N	Non-PEX Mean	PEX Mean	Mean Δ (%)	Non-PEX Median	PEX Median	Median Δ (%)	*p*-Value
Age	55	12	72.18	74.58	2.4 (3%)	74	75.5	1.5 (2%)	0.1465
DM duration	55	12	13.69	14.67	1 (7%)	12	14.5	2.5 (21%)	0.2749
DR stage	55	12	1.82	1.58	−0.2 (−13%)	2	1	−1 (−50%)	0.2281
**BCVA baseline**	**55**	**12**	**0.35**	**0.25**	**−0.1 (−29%)**	**0.3**	**0.3**	**0 (0%)**	**0.0455**
BCVA follow-up	55	12	0.73	0.68	0 (−7%)	0.8	0.65	−0.2 (−19%)	0.3539
FAZ baseline	55	12	0.29	0.31	0 (7%)	0.29	0.28	0 (−3%)	0.2294
FAZ follow-up	47	12	0.29	0.29	0 (0%)	0.28	0.28	0 (0%)	0.4850
FAZ PER baseline	54	12	2.23	2.16	−0.1 (−3%)	2.2	2.27	0.1 (3%)	0.3604
FAZ PER follow-up	47	12	2.19	2.1	−0.1 (−4%)	2.12	2.25	0.1 (6%)	0.4588
PAR SCP baseline	55	12	41.44	42.69	1.3 (3%)	40.9	41.2	0.3 (1%)	0.2088
PAR DCP baseline	55	12	45.93	43.73	−2.2 (−5%)	47.8	43.1	−4.7 (−10%)	0.1778
PAR SCP follow-up	46	12	43.77	40.92	−2.9 (−7%)	43.85	42.2	−1.7 (−4%)	0.0871
PAR DCP follow-up	47	12	48.61	47.11	−1.5 (−3%)	48.8	47.7	−1.1 (−2%)	0.2065
PER DCP baseline	54	12	38.51	37.36	−1.2 (−3%)	38.25	37.45	−0.8 (−2%)	0.2749
PER SCP baseline	54	12	42.12	41.42	−0.7 (−2%)	42.05	40.35	−1.7 (−4%)	0.3508
PER DCP follow-up	46	12	43.29	41.75	−1.5 (−4%)	44.5	39.6	−4.9 (−11%)	0.2282
PER SCP follow-up	46	12	43.15	42.51	−0.6 (−1%)	43.75	43.25	−0.5 (−1%)	0.3363
CRT baseline	55	12	271.56	300.67	29.1 (11%)	260	283	23 (9%)	0.1553
**CRT follow-up**	**48**	**12**	**282.9**	**324.08**	**41.2 (15%)**	**268.5**	**300**	**31.5 (12%)**	**0.0490**

Values for PEX and non-PEX subjects, as well as between-group comparisons at baseline and follow-up. All the significant results are shown in bold. Abbreviations: DM, diabetes mellitus; DR, diabetic retinopathy; BCVA, best-corrected visual acuity; FAZ, foveal avascular zone; PER, perimeter; PAR, parafoveal; PER, perifoveal; DCP, deep capillary plexus; SCP, superficial capillary plexus; CRT, central retinal thickness.

**Table 3 biomedicines-13-00908-t003:** Changes in initial and follow-up measurements within PEX and non-PEX groups.

PEX	Variable	n	Mean Before	Mean After	Mean Δ (%)	Median Before	Median After	Median Δ (%)	*p*-Value
no	BCVA	55	0.35	0.73	0.38 (108%)	0.30	0.80	0.5 (167%)	<0.0001
yes	BCVA	12	0.25	0.68	0.43 (174%)	0.30	0.65	0.35 (117%)	0.0019
no	FAZ	47	0.29	0.29	0 (0%)	0.29	0.28	−0.01 (−2%)	0.4621
yes	FAZ	12	0.31	0.29	−0.02 (−5%)	0.28	0.28	0 (1%)	0.8833
no	FAZPER	47	2.23	2.19	−0.04 (−2%)	2.20	2.13	−0.07 (−3%)	0.9251
yes	FAZPER	12	2.16	2.10	−0.06 (−3%)	2.27	2.25	−0.02 (−1%)	0.8982
no	PARSCP	46	41.44	43.77	2.33 (6%)	40.90	43.85	2.95 (7%)	0.0034
yes	PARSCP	12	42.69	40.93	−1.77 (−4%)	41.20	42.20	1 (2%)	0.8706
no	PARDCP	47	45.93	48.61	2.67 (6%)	47.80	48.80	1 (2%)	0.0104
yes	PARDCP	12	43.73	47.11	3.38 (8%)	43.10	47.70	4.6 (11%)	0.0517
no	PERDCP	46	38.51	43.29	4.78 (12%)	38.25	44.50	6.25 (16%)	<0.0001
yes	PERDCP	12	37.36	41.75	4.39 (12%)	37.45	39.60	2.15 (6%)	0.0037
no	PERSCP	46	42.12	43.15	1.02 (2%)	42.05	43.75	1.7 (4%)	0.0759
yes	PERSCP	12	41.43	42.51	1.08 (3%)	40.35	43.25	2.9 (7%)	0.2479
no	CRT	48	271.56	282.90	11.33 (4%)	260.00	268.50	8.5 (3%)	<0.0001
yes	CRT	12	300.67	324.08	23.42 (8%)	283.00	300.00	17 (6%)	0.0126

## Data Availability

The data supporting the findings of this study are available from the corresponding author upon reasonable request.

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
