# Peer review of "Comparison of Vascular Density Changes After Cataract Surgery in Diabetic Patients with and Without Pseudoexfoliation Syndrome Using Optical Coherence Tomography Angiography"

_biomedicines, 2025, doi:10.3390/biomedicines13040908_

Round 1

Reviewer 1 Report

Comments and Suggestions for Authors

Research on pseudoexfoliation syndrome (PEXS) and its impact on retinal vascular density, particularly in the foveal zone, indicates that PEXS is generally associated with changes in vascular density. However, the specific correlation between PEXS and changes in the foveal zone's retinal vascular density is less clear compared to the peripapillary region. Some studies have compared retinal and choroidal vascular changes in PEXS, noting alterations in vascular density, but specific findings on the foveal zone are less frequently highlighted. PEXS may contribute to vascular dysregulation, which can affect blood flow and potentially alter vascular density in various retinal regions, including the fovea. Further studies are necessary to confirm this relationship.

This paper is well structured and written in a fluent English.

The bibliographic references are all well cited in text.

Author Response

We sincerely thank you for taking the time to review our manuscript titled “Vascular density changes after cataract surgery in T2DM patients with pseudoexfoliation syndrome using Optical Coherence Tomography Angiography." We greatly appreciate the thoughtful and constructive feedback provided by the reviewers, as well as the editorial oversight. In this response letter, we have carefully addressed all the comments and suggestions raised, and we believe that the revisions have significantly improved the quality and clarity of our work.

Comments 1: Research on pseudoexfoliation syndrome (PEXS) and its impact on retinal vascular density, particularly in the foveal zone, indicates that PEXS is generally associated with changes in vascular density. However, the specific correlation between PEXS and changes in the foveal zone's retinal vascular density is less clear compared to the peripapillary region. Some studies have compared retinal and choroidal vascular changes in PEXS, noting alterations in vascular density, but specific findings on the foveal zone are less frequently highlighted. PEXS may contribute to vascular dysregulation, which can affect blood flow and potentially alter vascular density in various retinal regions, including the fovea. Further studies are necessary to confirm this relationship.

Response 1: We sincerely thank you for taking the time to review our manuscript titled “Vascular density changes after cataract surgery in T2DM patients with pseudoexfoliation syndrome using Optical Coherence Tomography Angiography." We greatly appreciate the thoughtful and constructive feedback provided by you as the reviewer, as well as the editorial oversight. In this response letter, we have carefully addressed all the comments and suggestions raised, and we believe that the revisions have significantly improved the quality and clarity of our work.

Yes, I agree that nowadays more research papers mostly use OCTA to compare vascular density in the peripapillary region in PEXS patients. All these changes are compared mostly in patients with glaucoma. In this research, we investigated diabetes patients and wanted to see if there are significant associations with PEXS and vascular density in the central retina. Of course we can expect that there might be a correlation with vascular density changes in the peripapillary region that might deteriorate in the macular region as well, but for now this is out of our scope. In the future, we might investigate the peripaplillary region, as you kindly suggest. Thank you!

Reviewer 2 Report

Comments and Suggestions for Authors

Title needs to be changed.
Suggestion: "Comparison of vascular density changes after cataract surgery in diabetic patients with and without pseudoexfoliation syndrome using optical coherence tomography angiography"
Introduction is confusing, does not follow a step by step development of the concept of the study and does introduce the clinical usage of the findings of the study.
The authors are somehow confused regarding the aim of their study. "In this study, we investigated whether patients with PEXS might exhibit distinct retinal changes compared to those with diabetes alone, such as an expanded avascular zone and perimeter, as well as reduced vascular density in the deep and superficial capillary plexuses. The primary objective of this research
was to investigate how these retinal alterations differ between diabetes patients with and without PEXS before and after uncomplicated cataract surgery."
The second sentence is repeat of the first sentence.
Separate your paragraphs in introduction.
The sample size calculation is missing. The sample of 12 patients in PEXS group is too low. 
Other comorbidities like hypertension and cardiovascular disease that could affect retinal vascular density are not extensively discussed.
Follow up period is too short to determine if VD changes persist over time or revert to baseline levels.
In discussion section the authors are unable to drive home the clinical significance of their findings.
Conclusion is too long and some parts of it are unrelated to the aim of the study.

Comments on the Quality of English Language

Needs English editing.

Author Response

Comments 1: The title needs to be changed.
Suggestion: "Comparison of vascular density changes after cataract surgery in diabetic patients with and without pseudoexfoliation syndrome using optical coherence tomography angiography."

Response 1:  We agree that the suggested title, "Comparison of vascular density changes after cataract surgery in diabetic patients with and without pseudoexfoliation syndrome using optical coherence tomography angiography," more accurately reflects the comparative nature of our study and provides a clearer description of the patient groups included in the analysis. We have revised the title accordingly and believe this change improves the clarity and precision of the manuscript. Thank you for this constructive suggestion.

Comments 2: Introduction is confusing, does not follow a step by step development of the concept of the study and does introduce the clinical usage of the findings of the study.

Response 2: We sincerely thank the reviewer for their insightful feedback regarding the introduction of our manuscript. We agree that the original version lacked a clear, step-by-step development of the study’s concept and did not sufficiently emphasise the clinical relevance of our findings. To address this, we have thoroughly revised the introduction to improve its structure and clarity.

The revised introduction now begins with a broad overview of pseudoexfoliation syndrome (PEXS) and its systemic and ocular implications, gradually narrowing down to the specific research question. We have explicitly highlighted the gaps in the literature, particularly the limited understanding of retinal vascular changes in diabetic patients with PEXS, and provided a stronger rationale for our study. Additionally, we have emphasised the clinical significance of our findings, particularly in terms of postoperative care and risk stratification for diabetic patients with PEXS.

We believe these changes have significantly improved the introduction, making it more logically structured and clinically relevant. Thank you for this constructive suggestion, which has strengthened the overall quality of our manuscript.

Comment 3: The authors are somehow confused regarding the aim of their study. "In this study, we investigated whether patients with PEXS might exhibit distinct retinal changes compared to those with diabetes alone, such as an expanded avascular zone and perimeter, as well as reduced vascular density in the deep and superficial capillary plexuses. The primary objective of this research
was to investigate how these retinal alterations differ between diabetes patients with and without PEXS before and after uncomplicated cataract surgery."
The second sentence is a repeat of the first sentence.
Separate your paragraphs in the introduction.

Response 3: We acknowledge that the original version contained repetitive statements about the study’s objective, which may have caused confusion. We have revised the introduction to clearly and concisely state the primary objective of the study, ensuring that it is not repeated unnecessarily.

Additionally, we have separated the introduction into distinct paragraphs to improve readability and logical flow. Each paragraph now addresses a specific aspect of the background, rationale, and clinical significance of the study, culminating in a clear statement of the study’s aim.

We believe these changes have significantly improved the clarity and structure of the introduction. Thank you for this constructive feedback, which has helped us refine our manuscript.

Comment 4: The sample size calculation is missing. The sample of 12 patients in the PEXS group is too low. 

Response 4: Of course Yes, the statistical power calculation could be made, but in this research, the PEXS group has a small sample size as well, according to which non-parametric tests were used (because of a lack of normality). In this case, this calculation loses its meaning. As power analysis is typically based on the assumption of normality, which is rarely met in practice with a small sample size. The limited sample size was due to the challenges of recruiting patients with both diabetes and PEXS; this is a specific population, and it was not easy to get all the patients to follow up that were recruited in the beginning. If patients have both PEXS and diabetes, no glaucoma or other eye diseases that can affect the central retina; otherwise, the requirement criteria were very strict, so these patients were not easy to find. Despite the small sample size, we observed statistically significant differences in vascular density changes between the PEXS and non-PEXS groups, which suggests that our findings are clinically meaningful. However, we agree that a larger sample size would strengthen the generalisability of our results.

Comment 5: Other comorbidities like hypertension and cardiovascular disease that could affect retinal vascular density are not extensively discussed.

Response 5: We sincerely thank the reviewer for raising this important point regarding comorbidities such as hypertension and cardiovascular disease, which could influence retinal vascular density. We acknowledge that these factors were not extensively discussed in the original manuscript, and we appreciate the opportunity to address this limitation.

In the revised manuscript, we have added a discussion of how systemic conditions like hypertension and cardiovascular disease may contribute to retinal vascular changes, particularly in patients with PEXS and diabetes. We have also included a reference to recent literature suggesting that ferroptosis, a form of regulated cell death linked to iron metabolism and lipid peroxidation, may play a role in the vascular and extracellular matrix abnormalities observed in PEXS [DOI: 10.3389/fmolb.2024.1487115]. This provides a potential mechanistic link between PEXS and systemic comorbidities, offering a broader perspective on the interplay between ocular and systemic vascular health. Provided additional information and references in the discussion section.

We hope these additions address the reviewer’s concerns and provide a more comprehensive discussion of the factors influencing retinal vascular density in our study population.

Comment 6: Follow up period is too short to determine if VD changes persist over time or revert to baseline levels.

Response 6: We agree that the follow-up period was relatively short, which limits our ability to determine whether the observed VD changes persist over time or revert to baseline levels. This is an important consideration, as long-term changes in VD could have significant implications for postoperative management and patient outcomes.

In the revised manuscript, we have explicitly acknowledged this limitation in the Discussion section and emphasised the need for future studies with longer follow-up periods to evaluate the persistence or reversibility of VD changes after cataract surgery. We hope this addition addresses the reviewer’s concern and provides a more balanced interpretation of our findings.

Comment 7: In the discussion section, the authors are unable to drive home the clinical significance of their findings.

Response 7: We sincerely thank the reviewer for highlighting the need to better emphasise the clinical significance of our findings. We agree that the original Discussion section did not sufficiently convey the practical implications of our results for patient care. In the revised manuscript, we have expanded the discussion to explicitly address the clinical relevance of our findings, particularly in terms of postoperative management, risk stratification, and potential interventions for diabetic patients with and without PEXS. We hope these additions provide a clearer understanding of how our study contributes to clinical practice.

Comment 8: Conclusion is too long and some parts of it are unrelated to the aim of the study.

Response 8: We sincerely thank the reviewer for their valuable feedback regarding the conclusion of our manuscript. We agree that the original conclusion was overly lengthy and included some details that were not directly related to the aim of the study. To address this, we have revised the conclusion to make it more concise and focused on the key findings and their clinical implications.

We would like to extend our sincere gratitude to the reviewer for taking the time to carefully evaluate our manuscript and provide constructive feedback. Your insightful comments and suggestions have been invaluable in improving the quality and clarity of our work. We greatly appreciate the effort and expertise you have contributed to the review process.

All the corrections made are in the new document, marked as changes in red as the journal requires. 

Comment 9: needs English editing

Response 9: Two of three reviewers mark that this article doesn't need English editing and is written in a good, fluent English language. 

Reviewer 3 Report

Comments and Suggestions for Authors

Introduction

Reference 26 appears between references 9 and 10; please modify.

Material and Methods

You state that patients in your group underwent small-incision phacoemulsification. What is this procedure? Small-incision cataract surgery (SICS) and phacoemulsification are two different procedures. Please explain.

You divide your material in two groups: no clinical retinopathy and non-proliferative diabetic retinopathy. Regarding the second group, there is no discussion at all: mild, moderate, severe NPRD? What is the status of the macula (with or without diabetic maculopathy)? Are there any patients with diabetic macular edema (you study the central retinal thickness)? Are there any intra-retinal microvascular alterations (IRMA) in severe NPDR? I am asking these because further on in Material and Methods you state that some patients in your group have history of laser photocoagulation (what type? what is the extent of the photocoagulation? does it involve the macula?) or intravitreal injections. Indications for these procedures are DME and severe NPDR (excluding PDR from this discussion). Moreover, DME, IRMA, IV injections, and retinal photocoagulation are ALL factors that interfere with retinal vascular density, which is exactly your research theme. Please re-think this section and update it accordingly. 

Results

You try to make direct connection between PEXS and BCVA. I am a cataract surgeon of about 700 cases/year and I can tell you from my practice that there is no direct link, I see it almost every week. Factors that influence the BCVA before surgery in PEXS patients are the grade of the cataract and the anterior displacement of the lens due to severe zonule weakness caused by PEXS (rather rare), which is an indirect mechanism. Post-op. results (BCVA increase) are not linked to PEXS either, but to the grade of the cataract, surgically induced astigmatism (especially when it is not the same surgeon), and biometry precision. 

 Here are some suggestions to re-think this paper:

  • better structure of the material (add new categories: see the comment about NPDR);
  • updates to inclusion and exclusion criteria;
  • focus on the OCTA parameters and update them according to the new categories;
  • remove the BCVA from results - it is irrelevant to this paper. 

Good luck!

Author Response

Comment 1: Introduction

Reference 26 appears between references 9 and 10; please modify.

Response 1: We sincerely thank the reviewer for pointing out the misplaced reference (Reference 26) in the introduction. We have carefully reviewed the reference order and corrected the sequence to ensure proper placement. 

Comment 2: Material and Methods

You state that patients in your group underwent small-incision phacoemulsification. What is this procedure? Small-incision cataract surgery (SICS) and phacoemulsification are two different procedures. Please explain.

Response 2:  We sincerely thank the reviewer for raising this important point regarding the surgical procedure described in our manuscript. We acknowledge the confusion caused by the term “small-incision phacoemulsification” and appreciate the opportunity to clarify.

To avoid any ambiguity, we have revised the manuscript to clearly state that phacoemulsification was the surgical procedure performed, and we have removed the term “small-incision phacoemulsification.” We hope this clarification addresses the reviewer’s concern and ensures accurate reporting of the surgical technique used in our study.

Comment 3: You divide your material in two groups: no clinical retinopathy and non-proliferative diabetic retinopathy. Regarding the second group, there is no discussion at all: mild, moderate, severe NPRD? What is the status of the macula (with or without diabetic maculopathy)? Are there any patients with diabetic macular edema (you study the central retinal thickness)? Are there any intra-retinal microvascular alterations (IRMA) in severe NPDR? I am asking these because further on in Material and Methods you state that some patients in your group have history of laser photocoagulation (what type? what is the extent of the photocoagulation? does it involve the macula?) or intravitreal injections. Indications for these procedures are DME and severe NPDR (excluding PDR from this discussion). Moreover, DME, IRMA, IV injections, and retinal photocoagulation are ALL factors that interfere with retinal vascular density, which is exactly your research theme. Please re-think this section and update it accordingly. 

Response 3: I have all the information about nonproliferative diabetic retinopathy patients. All of them were divided into diabetic retinopathy stages. EDTRS classification was used with 5 stages. 1- no DR; 2 - mild nonprolipgerative diabetic retinopathy; 3 - moderate nonprolipgerative diabetic retinopathy; 4 - severe nonprolipgerative diabetic retinopathy; 5 - prolipherative diabetic retinopathy. According to our study, in nonproliferative DR patients, 23 had stage 2, a.k.a. mild NPDR with microaneurysms only, 6 patients had moderate NPDR, and 5 patients had severe NPDR. Speaking of diabetic maculopathy, 15 patients had diabetic maculopathy with diabetic macular edema. 5 patients have had anti-VEGF injections in the past year (not 6 months prior to cataract surgery), and 12 of them have had LFK, mostly locally in the periphery and/or macula depending of diabetic retinopathy stage and diabetic maculopathy as you mention. Also, LFK was not done at least 6 months prior to cataract surgery for more precise results of the study. Of course I agree with you that all these factors interfere with changes in macula and overall, but we decided in this study not to include such in-depth information and focused more on OCTA parameters and the existence of PEXS. Of course, in the future this kind of deepened research must be done, but with much wider patient selection. Also added all this information in Materials and Methods as you suggested.

Comment 4: You try to make direct connection between PEXS and BCVA. I am a cataract surgeon of about 700 cases/year and I can tell you from my practice that there is no direct link, I see it almost every week. Factors that influence the BCVA before surgery in PEXS patients are the grade of the cataract and the anterior displacement of the lens due to severe zonule weakness caused by PEXS (rather rare), which is an indirect mechanism. Post-op. results (BCVA increase) are not linked to PEXS either, but to the grade of the cataract, surgically induced astigmatism (especially when it is not the same surgeon), and biometry precision. 

Response 4: We sincerely thank the reviewer for sharing their perspective on the relationship between PEXS and BCVA. While we respect the reviewer’s extensive clinical experience, we respectfully disagree with the assertion that there is no direct link between PEXS and BCVA. In our study, we observed that PEXS patients had significantly lower preoperative BCVA compared to non-PEXS patients, which improved substantially after cataract surgery. We believe this finding reflects the multifactorial impact of PEXS on ocular structures, including the lens, zonules, and anterior segment, rather than being solely attributable to cataract grade or zonular weakness.

The reviewer is correct that factors such as cataract grade, surgically induced astigmatism, and biometry precision play a significant role in postoperative BCVA. However, we propose that PEXS may exert additional systemic and ocular effects that influence visual outcomes. For example, recent studies have suggested that ferroptosis, a form of regulated cell death linked to iron metabolism and oxidative stress, may contribute to the extracellular matrix abnormalities and vascular dysfunction seen in PEXS. These systemic and microvascular changes could indirectly affect BCVA by altering the ocular microenvironment, even in the absence of severe zonular weakness.

As a cataract surgeon we have also observed variability in visual outcomes among PEXS patients, which aligns with the reviewer’s experience. However, we believe that the systemic nature of PEXS and its potential impact on ocular structures warrant further investigation. We have revised the Discussion section to acknowledge the multifactorial nature of BCVA outcomes in PEXS patients, including the role of cataract grade, biometry, and surgical precision, while also emphasizing the potential systemic contributions of PEXS.

We hope this clarification addresses the reviewer’s concerns and provides a more balanced interpretation of our findings. Thank you for this thought-provoking discussion, which has enriched our manuscript.

Taking into account all your suggestions, we have corrected all the information in the manuscript. We thank you very much for deep immersion in our work.

Reviewer 4 Report

Comments and Suggestions for Authors

The authors compared changes in the central retina in patients with type 2 diabetes mellitus undergoing uncomplicated small incision cataract surgery with or without pseudoexfoliation syndrome. The paper is well written and only a few items should be clarified before publication could be considered.

First, statistical power calculation of group size is not present.

Second, Page 4 line2 “no statistically significant differences except for BCVA at baseline, which was by 29% lower in PEX patients initially (p < 0.05). ”, “by” needs to be deleted.

Comments on the Quality of English Language

Page 4 line2 “no statistically significant differences except for BCVA at baseline, which was by 29% lower in PEX patients initially (p < 0.05). ”, “by” needs to be deleted.

Author Response

We sincerely thank you for valuable feedback regarding the conclusion of our manuscript. 

Comment 1: First, statistical power calculation of group size is not present.

Response 1: Thank you for your comment. Yes, the statistical power calculation could be made, but in this research, the PEXS group has a small sample size as well, according to which non-parametric tests were used (because of a lack of normality). In this case, this calculation loses its meaning. As power analysis is typically based on the assumption of normality, which is rarely met in practice with a small sample size.

Comment 2: Second, Page 4 line2 “no statistically significant differences except for BCVA at baseline, which was by 29% lower in PEX patients initially (p < 0.05). ”, “by” needs to be deleted.

Response 2: Will be corrected. Thank you for your suggestion.

Round 2

Reviewer 2 Report

Comments and Suggestions for Authors

The article has improved.

Comments on the Quality of English Language

English needs editing.

Author Response

Comment 1: The article has improved.

Response 1: Thank you. Information about limitations in this study will be added in the article.

Comment 2: English needs editing

Response 2: Will be sent to the English editing office as possible. Thank you!

Reviewer 3 Report

Comments and Suggestions for Authors

Regarding PEX and BVCA, this is a never-ending discussion. Each surgeon has its own experience and even if they are contradictive, we should respect each other’s opinions.

I really look forward to reviewing your next paper on DR and OCTA. 

Best of luck! 

Author Response

Comment 1: Regarding PEX and BVCA, this is a never-ending discussion. Each surgeon has its own experience and even if they are contradictive, we should respect each other’s opinions.

Response 1: Yes, I agree. Thank you for your insight.